# Efficiently Identifying Task Groupings for Multi-Task Learning

**Christopher Fifty[1], Ehsan Amid[1], Zhe Zhao[1], Tianhe Yu[1,2],**
**Rohan Anil[1], Chelsea Finn[1,2]**
Google Research, Brain Team[1], Stanford University[2]
cfifty@google.com

## Abstract

Multi-task learning can leverage information learned by one task to benefit the training of other tasks. Despite this capacity, naïvely training all tasks together in one model often degrades performance, and exhaustively searching through combinations of task groupings can be prohibitively expensive. As a result, efficiently identifying the tasks that would benefit from training together remains a challenging design question without a clear solution. In this paper, we suggest an approach to select which tasks should train together in multi-task learning models. Our method determines task groupings in a single run by training all tasks together and quantifying the effect to which one task's gradient would affect another task's loss. On the large-scale Taskonomy computer vision dataset, we find this method can decrease test loss by 10.0% compared to simply training all tasks together while operating 11.6 times faster than a state-of-the-art task grouping method.

## 1  Introduction

Many of the forefront challenges in applied machine learning demand that a single model performs well on multiple tasks, or optimizes multiple objectives while simultaneously adhering to unmovable inference-time constraints. For instance, autonomous vehicles necessitate low inference time latency to make multiple predictions on a real-time video feed to precipitate a driving action [27]. Robotic arms are asked to concurrently learn how to pick, place, cover, align, and rearrange various objects to improve learning efficiency [25], and online movie recommendation systems model multiple engagement metrics to facilitate low-latency personalized recommendations [13]. Each of the above applications depends on multi-task learning, and advances which improve multi-task learning performance have the potential to make an outsized impact on these and many other domains.

Multi-task learning can improve modeling performance by introducing an inductive bias to prefer hypothesis classes that explain multiple objectives and by focusing attention on relevant features [43]. However, it may also lead to severely degraded performance when tasks compete for model capacity or are unable to build a shared representation that can generalize to all objectives. Accordingly, finding groups of tasks that derive benefit from the positives of training together while mitigating the negatives often improves the modeling performance of multi-task learning systems.

While recent work has developed new multi-task learning optimization schemes [28, 10, 45, 53, 11, 50], the problem of deciding which tasks should be trained together in the first place is an understudied and complex issue that is often left to human experts [56]. However, a human's understanding of similarity is motivated by their intuition and experience rather than a prescient knowledge of the underlying structures learned by a neural network. To further complicate matters, the benefit or detriment induced from multi-task learning relies on many non-trivial decisions including, but not limited to, dataset characteristics, model architecture, hyperparameters, capacity, and convergence [51, 49, 46, 47]. As a result, a systematic technique to determine which tasks

35th Conference on Neural Information Processing Systems (NeurIPS 2021).

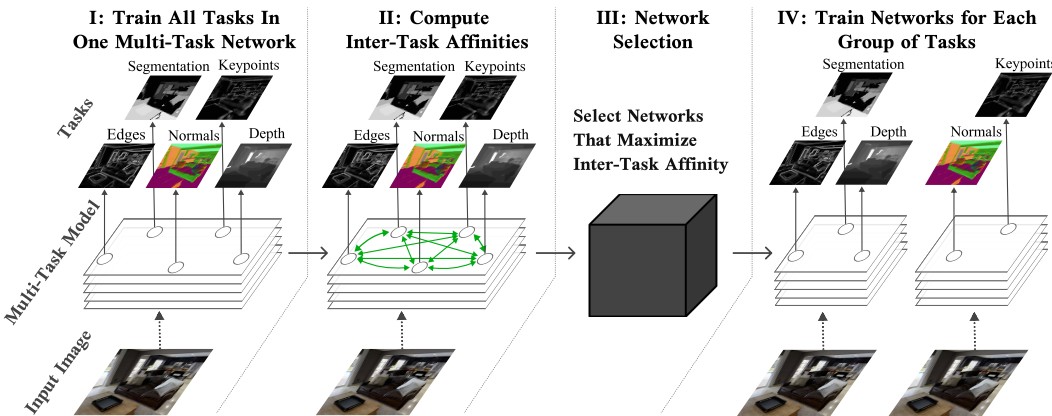

Figure 1: Overview of our suggested approach to efficiently determine task groupings. (I): Train all tasks together in a multi-task learning model. (II): Compute inter-task affinity scores during training. (III): Select multi-task networks that maximize the inter-task affinity score onto each serving-time task. (IV): Train the resulting networks and deploy to inference.

should train together in a multi-task neural network would be valuable to practitioners and researchers alike [5, 6].

One approach to select task groupings is to exhaustively search over the $2^{|\mathcal{T}|} - 1$ multi-task networks[1] for a set of tasks $\mathcal{T}$. However, the cost associated with this search can be prohibitive, especially when there is a large number of tasks. It is further complicated by the fact that the set of tasks to which a model is applied may change throughout its lifetime. As tasks are added to or dropped from the set of all tasks, this costly analysis would need to be repeated to determine new groupings. Moreover, as model scale and complexity continues to increase, even approximate task grouping algorithms which evaluate only a subset of combinations may become prohibitively costly and time-consuming to evaluate.

In this paper, we aim to develop an efficient framework to select task groupings without sacrificing performance. We propose to measure inter-task affinity by training all tasks together in a single multi-task network and quantifying the effect to which one task's gradient update would affect another task's loss. This per-step quantity is averaged across training, and tasks are then grouped together to maximize the affinity onto each task. A visual depiction of the method is shown in Figure 1. Our suggested approach makes no assumptions regarding model architecture and is applicable to any paradigm in which shared parameters are updated with respect to multiple losses.

In summary, our primary contribution is to suggest a measure of inter-task affinity that can be used to systematically and efficiently determine task groupings for multi-task learning. Our theoretical analysis shows that grouping tasks by maximizing inter-task affinity will outperform any other task grouping in the convex setting under mild conditions. Further on two challenging multi-task image benchmarks, our empirical analysis finds this approach outperforms training all tasks independently, training all tasks together (with and without training augmentations), and is competitive with a state-of-the-art task grouping method while decreasing runtime by more than an order of magnitude.

## 2 Related Work

**Task Groupings.** Prevailing wisdom suggests tasks which are similar or share a similar underlying structure may benefit from training together in a multi-task system [9, 8, 4]. Early work in this domain pertaining to the convex setting assume all tasks share a common latent feature representation, and find that model performance can be significantly improved by clustering tasks based on the basis vectors they share in this latent space [26, 30]. However, early convex methods to determine task groupings often make prohibitive assumptions that do not scale to deep neural networks.

Deciding which tasks should train together in multi-task neural networks has traditionally been addressed with costly cross-validation techniques or high variance human intuition. An altogether

---

[1]There are $\binom{|\mathcal{T}|}{1}$ single-task networks, $\binom{|\mathcal{T}|}{2}$ 2-task multi-task networks, ... which yields $\sum_{i=1}^{|\mathcal{T}|} \binom{|\mathcal{T}|}{i} = 2^{|\mathcal{T}|} - 1$.

different approach may leverage recent advances in transfer learning focused on understanding task relationships [54, 3, 15, 58, 2]; however, [46] show transfer learning algorithms which determine task similarity do not carry over to the multi-task learning domain and instead propose a multi-task specific framework which trains between $\binom{|\mathcal{T}|}{2} + |\mathcal{T}|$ and $2^{|\mathcal{T}|} - 1$ models to approximate exhaustive search performance. Our approach differs from [46] in that it computes task groupings from only a single training run.

**Architectures and Training Dynamics.** A plethora of multi-task methods addressing what parameters to share among tasks in a model have been developed, such as Neural Architecture Search [19, 47, 49, 35, 22, 39], Soft-Parameter Sharing [40, 14, 52], and asymmetric information transfer [31, 44, 32] to improve multi-task performance. Although this direction is promising, we direct our focus towards *when to share* tasks in a multi-task network rather than architecture modifications to maximize the benefits from training a fixed set of tasks together. Nevertheless, both approaches are complementary, and architecture augmentations seem to perform best when trained with related tasks [43].

Significant effort has also been invested to improve the optimization dynamics of MTL systems. In particular, dynamic loss reweighing has achieved performance superior to using fixed loss weights found with extensive hyperparameter search [28, 18, 37, 10, 45, 33]. Another set of methods seek to mitigate inter-task conflict by manipulating the direction of task gradients rather than simply their magnitude [48, 57, 53, 11, 50, 36]. In our experiments, we compare against Uncertainty Weights [28], GradNorm [10], and PCGrad [53] to contextualize the relative change in performance from splitting tasks into groups. We find that task grouping methods outperform all three training augmentations; nonetheless, they are naturally complementary. In Section 5, our results indicate enhancing the networks found by our method with PCGrad can lead to additional improvements in performance.

**Looking into the Future.** "Lookahead" methods in deep learning can often be characterized by saving the current state of the model, applying one or more gradient updates to a subset of the parameters, reloading the saved state, and then leveraging the information learned from the future state to modify the current set of parameters. This approach has been used extensively in the meta-learning [16, 42, 7, 17, 29], optimization [41, 21, 55, 23, 24], and recently auxiliary task learning domains [34]. Unlike the above mentioned methods which look into the future to modify optimization processes, our work adapts this central concept to the multi-task learning domain to characterize task interactions and assign tasks to groups of networks.

# 3 Task Grouping Problem Definition

We draw a distinction between inference-time latency constraints and inference-time memory budget. The former characterizes the speed at which predictions can be computed, with similarly sized models running in parallel having latency roughly equivalent to a single model running by itself. The latter relates to the number of parameters used by all models during inference, with a $n$-times parameter model having a similar budget to an $n$-group of normal-sized models. We configure our analysis to span both dimensions, but also provide analysis into only the latter in the Appendix.

Given a set of tasks $\mathcal{T}$, a fixed inference-time memory budget $b$, and latency constraint $c$, our aim is to assign tasks to networks such that average task performance is maximized. Additionally, each network must have a parameter count less than $c$; the networks must span our set of tasks $\mathcal{T}$; and the total number of networks is less than or equal to our memory budget $b$. Moreover, tasks can be trained in a network without being served from this network during inference. In this case, they are used to assist the learning of the other tasks, and during inference, are served from one of the other multi-task networks. Formulating our task grouping framework in this manner aligns our work with industry trends where inference-time parameter budgets are limited and latency unmovable.

More formally, for a set of $n$ tasks $\mathcal{T} = \{\tau_1, \tau_2, .., \tau_n\}$, we would like to construct a group of $k$ multi-task neural networks $M = \{m_1, m_2, ..., m_k\}$ such that $\forall \tau_i \in \mathcal{T}$, $\exists$ **exactly one** multi-task network $m_j \in M$, parameter count of $m_j < c$, such that $m_j$ makes an inference-time prediction for $t_i$ subject to $k \leq b$ where $b$ is our memory budget. $m_j$ is the $j^{th}$ multi-task network which takes input $\mathcal{X}$, and concurrently trains a set of tasks $\{\tau_a, \tau_c, ..., \tau_f\}$, but only serves a subset of those tasks at inference. For a given performance measure $\mathcal{P}$, we can then define the aggregate performance of our task grouping as $\sum_{i=1}^{n} \mathcal{P}(\tau_i|M)$ where $\mathcal{P}(\tau_i|M)$ computes the performance of task $\tau_i$ from the

set of models $M$ using the model $m_j$ which the task grouping algorithm predicts the performance of $\tau_i$ will be highest.

# 4 Grouping Tasks by Measuring Inter-Task Affinity

We propose a method to group tasks by examining the effect to which one task's gradient would increase or decrease another task's loss. We formally define the method in Section 4.1, describe a systematic procedure to go from inter-task affinity scores to a grouping of tasks in Section 4.2, and provide theoretical analysis in Section 4.3.

## 4.1 Inter-Task Affinity

Within the context of a hard-parameter sharing paradigm, tasks collaborate to build a shared feature representation which is then specialized by individual task-specific heads to output a prediction. Specifically, through the process of successive gradient updates to the shared parameters, tasks implicitly transfer information to each other. As a consequence, we propose to view the extent to which a task's successive gradient updates on the shared parameters affect the objective of other tasks in the network as a proxy measurement of inter-task affinity.

Consider a multitask loss function parameterized by $\{\theta_s\} \cup \{\theta_i \,|\, i \in \mathcal{T}\}$ where $\theta_s$ represents the shared parameters and $\theta_i$ represents the task $i \in \mathcal{T}$ specific parameters. Given a batch of examples $\mathcal{X}$, let

$$L_{\text{total}}(\mathcal{X}, \theta_s, \{\theta_i\}) = \sum_{i \in \mathcal{T}} L_i(\mathcal{X}, \theta_s, \theta_i)\,,$$

denote the total loss where $L_i$ represents the non-negative loss of task $i$. For simplicity of notation, we set the loss weight of each task to be equal to 1, though our construction generalizes to arbitrary weightings.

For a given training batch $\mathcal{X}^t$ at time-step $t$, define the quantity $\theta_{s|i}^{t+1}$ to represent the updated shared parameters after a gradient step with respect to the task $i$. Assuming stochastic gradient descent for simplicity, we have

$$\theta_{s|i}^{t+1} := \theta_s^t - \eta \nabla_{\theta_s^t} L_i(\mathcal{X}^t, \theta_s^t, \theta_i^t)\,.$$

We can now calculate a *lookahead* loss for each task by using the updated shared parameters while keeping the task-specific parameters as well as the input batch unchanged. That is, in order to assess the effect of the gradient update of task $i$ on a given task $j$, we can compare the loss of task $j$ before and after applying the gradient update from task $i$ onto the shared parameters. To eliminate the scale discrepancy among different task losses, we consider the ratio of a task's loss before and after the gradient step on the shared parameters as a scale invariant measure of relative progress. We can then define an asymmetric measure for calculating the *affinity* of task $i$ at a given time-step $t$ on task $j$ as

$$\mathcal{Z}_{i \to j}^t = 1 - \frac{L_j(\mathcal{X}^t, \theta_{s|i}^{t+1}, \theta_j^t)}{L_j(\mathcal{X}^t, \theta_s^t, \theta_j^t)}\,. \tag{1}$$

Notice that a positive value of $\mathcal{Z}_{i \to j}^t$ indicates that the update on the shared parameters results in a lower loss on task $j$ than the original parameter values, while a negative value of $\mathcal{Z}_{i \to j}^t$ indicates that the shared parameter update is antagonistic for this task's performance. Our suggested measure of inter-task affinity is computed at a per-step level of granularity, but can be averaged across all steps, every $n$ steps, or a contiguous subset of steps to derive a "training-level" score:

$$\hat{\mathcal{Z}}_{i \to j} = \frac{1}{T} \sum_{t=1}^{T} \mathcal{Z}_{i \to j}^t\,.$$

In Section 5, we find $\hat{\mathcal{Z}}_{i \to j}$ is empirically effective in selecting high performance task groupings and provide an ablation study along this dimension in Section 5.2.

## 4.2 Network Selection Algorithm

At a high level, our proposed approach trains all tasks together in one model, measures the pairwise task affinities throughout training, identifies task groupings that maximize total inter-task affinity,

and then trains the resulting groupings for evaluation on the test set. We denote this framework as Task Affinity Grouping (TAG) and now describe the algorithm to convert inter-task affinity scores into a set of multi-task networks. More formally, given $\binom{|\mathcal{T}|}{2}$ values representing the pairwise inter-task affinity scores collected during a single training run, our network selection algorithm should produce $k$ multi-task networks, $k \leq b$ where $b$ is the inference-time memory budget, with the added constraint that every task must be served from exactly one network at inference.

For a group composed of a pair of tasks $\{a, b\}$, the affinity score onto task $a$ would simply be $\hat{\mathcal{Z}}_{b \to a}$ and the affinity score onto task $b$ would be $\hat{\mathcal{Z}}_{a \to b}$. For task groupings consisting of three or more tasks, we approximate the inter-task affinity onto a given task by averaging the pairwise affinities onto this given task. Consider the group consisting of tasks $\{a, b, c\}$. We can compute the total inter-task affinity onto task $a$ by averaging the pair-wise affinities from tasks $b$ and $c$ onto $a$: $(\hat{z}_{b \to a} + \hat{z}_{c \to a})/2$.

After approximating higher-order affinity scores for each network consisting of three or more tasks, we select a set of $k$ multi-task networks such that the total affinity score onto each task used during inference is maximized. Informally, each task that is being served at inference should train with the tasks which most decrease its loss throughout training. This problem is NP-hard (reduction from Set-Cover) but can be solved efficiently with a branch-and-bound-like algorithm as detailed in [46] or with a binary integer programming solver as done by [54].

## 4.3  Theoretical Analysis

We now offer a theoretical analysis of our measure of inter-task affinity. Specifically, our goal is to provide an answer to the following question: given that task $b$ induces higher inter-task affinity than task $c$ on task $a$, does training $\{a, b\}$ together result in a lower loss on task $a$ than training $\{a, c\}$? Intuitively, we expect the answer to this question to always be positive. However, it is easy to construct counter examples for simple quadratic loss functions where training $\{a, b\}$ actually induces a higher loss than training $\{a, c\}$ (see Appendix). Nonetheless, we show that in a convex setting and under some mild assumptions, the grouping suggested by our measure of inter-task affinity must induce a lower loss value on task $a$.

For simplicity of notation, we ignore the task specific parameters and use $L_a(\theta)$ to denote the loss of task $a$ evaluated at shared parameters $\theta$. Additionally, we denote the gradient of tasks $a$, $b$, and $c$ at $\theta$ as $g_a$, $g_b$, and $g_c$, respectively.

**Lemma 1.** *Let $L_a$ be a $\alpha$-strongly convex and $\beta$-strongly smooth loss function. Given that task $b$ induces higher inter-task affinity than task $c$ on task $a$, the following inequality holds:*

$$g_a \cdot g_c - \frac{\beta \eta}{2} \|g_c\|^2 + \frac{\alpha \eta}{2} \|g_b\|^2 \leq g_a \cdot g_b \tag{2}$$

The proof is given in the Appendix.

**Proposition 1.** *Let $L_a$ be a $\alpha$-strongly convex and $\beta$-strongly smooth loss function. Let $\eta \leq \frac{1}{\beta}$ be the learning rate. Suppose that in a given step, task $b$ has higher inter-task affinity than task $c$ on task $a$. Moreover, suppose that the gradients have equal norm, i.e. $\|g_a\| = \|g_b\| = \|g_c\|$. Then, taking a gradient step on the parameters using the combined gradient of $a$ and $b$ reduces $L_a$ more so than taking a gradient step on the parameters using the combined gradient of $a$ and $c$, given that $\cos(g_a, g_c) \leq \frac{\eta}{4} \frac{\alpha \beta}{\beta - \alpha} - 1$ where $\cos(u, v) := \frac{u \cdot v}{\|u\| \|v\|}$ is the cosine similarity between $u$ and $v$.*

The proof is given in the Appendix. Proposition 1 intuitively implies the grouping chosen by maximizing per-task inter-task affinity is guaranteed to make more progress than any other group. Moreover, the assumptions for this condition to hold in the convex setting are fairly mild. Specifically, our first assumption relies on the gradients to have equal norms, but our proof can be generalized to when the gradient norms are approximately equal. This is often the case during training when tasks use similar loss functions and/or compute related quantities. The second assumption relies on the cosine similarity between the lower-affinity task $c$ and the primary task $a$ being smaller than a constant. This is a mild assumption since the ratio $\frac{\beta/\alpha}{\beta/\alpha - 1}$ becomes sufficiently large for the assumption to hold trivially when the Hessian of the loss has a sufficiently small condition number.

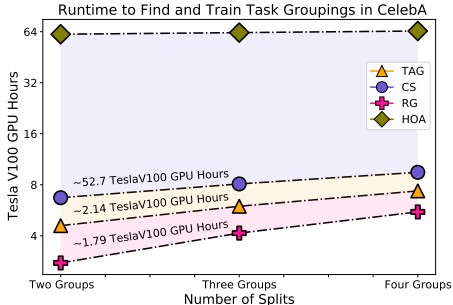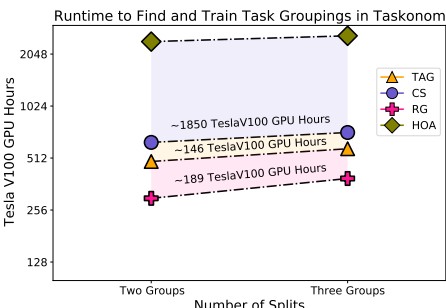

Figure 2: (Left) time to determine and train task groupings in CelebA. (Right) time to determine and train task groupings in Taskonomy. Note the **y-axis is in log scale**, and the time to determine task groupings is incurred only once to determine groupings for all splits.

# 5   Experiments

We evaluate[2] the capacity of TAG to select task groupings on CelebA, a large-scale face attributes dataset [38] and Taskonomy, a massive computer vision dataset of indoor scenes [54]. Following this analysis, we direct our focus towards answering the following questions with ablation experiments on CelebA:

- Does our measure of inter-task affinity align with identifying which tasks should train together?
- Should inter-task affinity be measured at every step of training to determine task groupings?
- Is measuring the change in train loss comparable with the change in validation loss when computing inter-task affinity?
- How do inter-task affinities change over the course of training?
- Do changes in a model's hyperparameters change which tasks should be trained together?

As described in Section 3, we constrain all networks to have the same number of parameters to adhere to a fixed inference-time latency constraint. We provide experimental results removing this constraint, as well as additional experimental results and detail relating to experimental design, in the Appendix.

## 5.1   Supervised Task Grouping Evaluation

For our task grouping evaluation, we compare two classes of approaches: approaches that determine task groupings, and approaches that train on all tasks together but alter the optimization. In the first class, we consider simply training all tasks together in the same network (MTL), training every task by itself (STL), the expected value from randomly selecting task groupings (RG), grouping tasks by maximizing inter-task cosine similarity between pairs of gradients (CS), our method (TAG ), and HOA [46] which approximates higher-order task groupings from pair-wise task performance. For the latter class, we consider Uncertainty Weights (UW) [28], GradNorm (GN) [10], and PCGrad [53]. In principle, these two classes of approaches are complementary and can be combined.

Our empirical findings are summarized in Figure 3. For the task grouping methods TAG , CS, and HOA, we report the time to determine task groupings, not determine task groupings and train the resultant multi-task networks. The runtime of RG is fixed to the time it takes to train a single multi-task network, and the runtime of STL is reported as the time it takes to train all single task networks. This is to facilitate comparison between the efficiency of different task grouping methods and provide a high-level overview of how long it takes to select task groupings compared to popular multi-task learning benchmarks. We also include a comparison of the time to determine task groupings and train the resultant multi-task networks among task grouping methods in Figure 2.

**CelebA.** We select a subset of 9 attributes {a1, a2, a3, a4, a5, a6, a7, a8, a9} from the 40 possible attributes in CelebA and optimize the baseline MTL model by tuning architecture, batch size, and

---

[2]Our code is available at github.com/google-research/google-research/tree/master/tag.

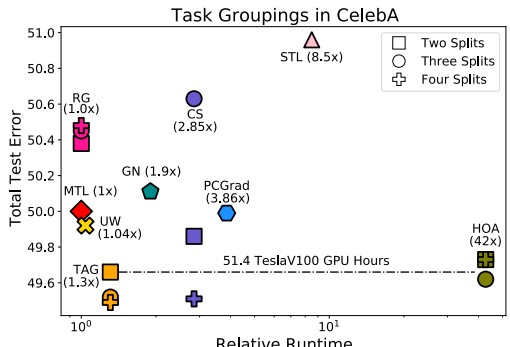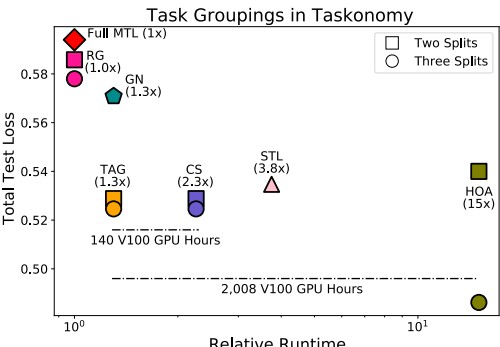

Figure 3: (Left) average classification error for 2, 3, and 4-split task groupings for the subset of 9 tasks in CelebA. (Right) total test loss for 2 and 3-split task groupings for the subset of 5 tasks in Taskonomy. All models were run on a TeslaV100 instance with the time to train the full MTL model being approximately 83 minutes in CelebA and 146 hours in Taskonomy. Note the x-axis is in log scale, and the relative runtime for TAG , CS, and HOA only considers the time to find groups.

learning rate to maximize the performance of training all tasks together on the validation set. We do not tune other methods with the exception of GradNorm, for which we search over {0.1, 0.5, 1.0, 1.5, 2.0, 3.0, 5.0} for alpha. For task grouping algorithms, we evaluate the set of {2-splits, 3-splits, 4-splits} inference-time memory budgets. Each of the multi-task networks has the same number of parameters as the base MTL model; however, the inference-time latency constraint is satisfied as all models may run in parallel. Our findings are summarized in Figure 3 (left). Alternatively, increasing the number of parameters within a single model may significantly increase the time to make a forward pass during inference. Figure 6 visualizes our findings when we remove the inference-time latency constraint from comparison systems.

We find the performance of TAG surpasses that of the HOA, RG, and CS task grouping methods, while operating 22 times faster than HOA. We also find UW, GN, and PCGrad to perform worse than the groups found by either HOA or TAG, suggesting the improvement from identifying tasks which train well together cannot be replaced with current multi-task training augmentations. Nevertheless, we find multi-task training augmentations can be complementary with task grouping methods. For example, augmenting the groups found by TAG with PCGrad improves 2-splits performance by 0.85%, 3-splits performance by 0.18%, but changes 4-splits performance by -0.08%.

Similar to the results from [46], we find GradNorm [10] can sometimes perform worse than training all tasks together. We reason this difference is due to our common experimental design that loads the weights from the epoch with lowest validation loss to reduce overfitting, and differs from prior work which uses model weights after training for 100 epochs [45]. Reformulating our design to mirror [45], we find training the model to 100 epochs without early stopping results in worse performance on MTL, GradNorm, UW, and PCGrad; however, with this change, each training augmentation method now significantly outperforms the baseline MTL method.

**Taskonomy.** Following the experimental setup of [46], we evaluate the capacity of TAG to select task groupings on the "Semantic Segmentation", "Depth Estimation", "Keypoint Detection", "Edge Detection", and "Surface Normal Prediction" objectives in Taskonomy. Unlike [46], our evaluation uses an augmented version of the medium Taskonomy split (2.4 TB) as opposed to the Full+ version (12 TB) to reduce computational overhead and increase reproducibility.

Our results are summarized in Figure 3 (right). Similar to our findings on CelebA, TAG continues to outperform MTL by 10.0%, GN by 7.7%, STL by 1.5%, and RG by 9.5%. Comparing TAG to HOA, we find the 2-split task grouping found by TAG to surpass the performance of that found by HOA by 2.5%, but HOA's 3-split task grouping performance is superior to that of TAG. In terms of compute, TAG is significantly more efficient than HOA, with HOA demanding an additional 2,008 TeslaV100 GPU hours to find task groupings. To put this cost into perspective, on an 8-GPU, on-demand p3.16xlarge AWS instance, the difference in monetary expenditure between TAG and HOA would be $6,144.48. On a similar note, the performance of TAG and CS on Taskonomy are equivalent, but TAG is more efficient, requiring 140 fewer TeslaV100 GPU hours to compute task groupings.

## 5.2 Multi-Task Ablation Studies

**Does our measure of inter-task affinity correlate with optimal task groupings?** To further evaluate if TAG can be used to select which tasks should train together in multi-task learning models, we evaluate its capacity to select the best training partner for a given task. We compare with the optimal (best) and worst auxiliary task computed from the test set and normalize scores with respect to the expected performance of selecting an auxiliary task at random.

Our findings are summarized in Table 1 and indicate the performance of auxiliary tasks found by TAG correlates with the performance of the optimal auxiliary task (Pearson's Correlation of 0.93%). However, a notable exception occurs with attribute a8 where TAG actually selects the worst partner. In this instance, and unlike every other objective in our dataset, no task manifests especially high or low inter-task affinity onto

| Tasks | Improvement in Test Accuracy Relative to Random Grouping | | |
|---|---|---|---|
| | optimal | (ours) | worst |
| a1 | 2.60% | 2.6% | -3.03% |
| a2 | 1.53% | 1.29% | -1.95% |
| a3 | 2.37% | 1.72% | -3.04% |
| a4 | 2.67% | 0.72% | -4.14% |
| a5 | 2.75% | 2.29% | -3.87% |
| a6 | 2.04% | 0.00% | -2.08% |
| a7 | 2.40% | 0.74% | -2.43% |
| a8 | 1.61% | -1.59% | -1.59% |
| a9 | 8.38% | 8.38% | -6.21% |

Table 1: Performance of each task when trained with it's partner task *relative* to the expected performance from random groupings.

a8. The difference in normalized inter-task affinity for a8 between the best and worst partner predicted by inter-task affinity is 0.04, while the next smallest difference is 4x larger at 0.16 for a3. A table showing these differences is included in the Appendix. This case represents a limitation of TAG, and it will struggle to find the best auxiliary task for objectives like a8.

However, this weakness does not seem to significantly affect the capacity of TAG to select strong task groupings. As no other task exhibits especially high or low inter-task affinity onto a8, it is often slotted into a larger task group rather than being one of the tasks with a large difference in inter-task affinity which precipitates the formation of a new group.

**Should inter-task affinity be computed at every step?**

It is likely that the inter-task affinity of consecutive steps are similar, and it could be the case that inter-task affinity signals at the beginning, middle, or end of training are sufficient for selecting which tasks should train together. In particular, if task relationships crystallize early in training, computing inter-task affinities during the initial stages of training may be sufficient. We evaluate both hypotheses on CelebA and our results are summarized in Table 2.

| Method | Relative Performance | Relative Speedup |
|---|---|---|
| Every 1 Step | 5.13% | 1.0x |
| Every 5 Steps | 5.13% | 2.56x |
| Every 10 Steps | 5.13% | 3.19x |
| Every 25 Steps | 4.84% | 3.73x |
| Every 50 Steps | 3.73% | 3.96x |
| Every 100 Steps | 2.06% | 4.08x |
| First 25% | 3.67% | 4.00x |
| Middle 25% | 4.31% | 2.95x |
| Final 25% | 4.02% | 2.34x |

Table 2: Change in total test accuracy across inference-time memory budget of {2-groups, 3-groups, 4-groups} relative to the expected performance from random groupings. Speedup is relative to computing inter-task affinity in each step.

Our findings indicate significant redundancy can be eliminated without degrading performance by computing inter-task affinities every 10 steps rather than every step. This modification increases training-time efficiency by 319% and is used in Section 5.1 to compute task groupings. After this threshold, signal strength decreases and leads to higher task grouping error. We also find that computing inter-task affinities in the first 25%, middle 25%, or final 25% of training degrades task-grouping performance. This result suggest the relationships among tasks change throughout training as measured by inter-task affinity. As a result, we choose to average inter-task affinity scores throughout the entirety of training to determine which tasks should train together.

**Is change in train loss comparable to change in validation loss?** Given multi-task learning's capacity to improve generalization, computing the change in validation loss after a gradient step may capture a more informative signal as to which tasks should train together. On the other hand, certain datasets may not contain a validation split and loading a batch from the validation set every 10 steps of training would decrease efficiency.

To our surprise, the inter-task affinity scores computed on the validation set are very similar to the inter-task affinity scores computed on the training set (Pearson's Coefficient: 0.9804). For context, this similarity with computing inter-task affinities every step is greater than any other ablation in Table 2 with the exception of "Every 5 Steps" as measured by Pearson's Coefficient. Moreover, the performance of groupings found by both methods are similar: 49.574 average total error across our

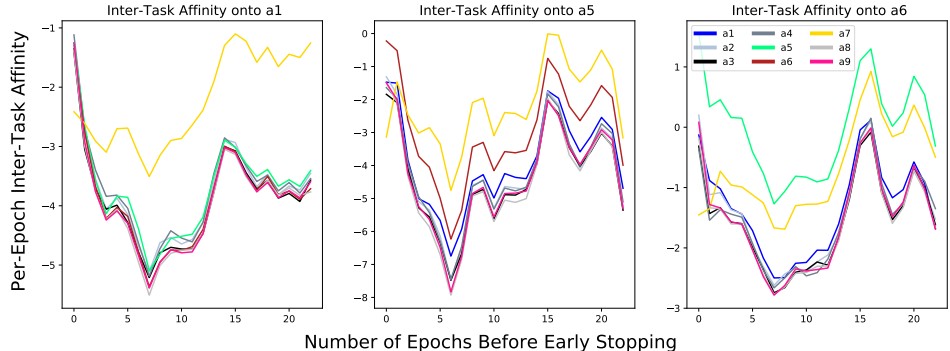

Figure 4: Total per-epoch inter-task affinities onto tasks {a1, a5, a6} in the CelebA dataset. The y-axis signifies the total change in train loss after applying an update to the shared parameters. Note the relationship among affinities changes. For instance in the center plot representing the inter-task affinity onto a5, a6 initially manifests higher affinity onto a5 than does a7. In later stages of training, this trend is reversed. In the same subplot, a1 manifests higher inter-task affinity over the tasks {a3, a4, a8, a9} at different stages of training.

inference time budget of {2-groups, 3-groups, 4-groups} compared with 49.576 for groupings found on the validation set.

More formally, let $X_{\mathrm{tr}}$ and $X_{\mathrm{val}}$ be independent and identically distributed random variables for training and validation, respectively. Given the updated shared parameter $\theta_{s|i}^{t+1}$ using the gradient of task $i$ calculated on $X_{\mathrm{tr}}$, the loss of task $j$ (s.t. $j \neq i$) yields similar expectations with respect to $X_{\mathrm{tr}}$ and $X_{\mathrm{val}}$. That is,

$$\mathbb{E}_{\mathrm{val}}[\mathbb{E}_{\mathrm{tr}}[\mathcal{L}_j(X_{\mathrm{val}}^t, \theta_{s|i}^{t+1}, \theta_j^t)]] \approx \mathbb{E}_{\mathrm{val}}[\mathcal{L}_j(X_{\mathrm{val}}^t, \mathbb{E}_{\mathrm{tr}}[\theta_{s|i}^{t+1}], \theta_j^t)] = \mathbb{E}_{\mathrm{tr}}[\mathcal{L}_j(X_{\mathrm{tr}}^t, \mathbb{E}_{\mathrm{tr}}[\theta_{s|i}^{t+1}], \theta_j^t)].$$

The leftmost term is a joint expectation w.r.t. both $X_{\mathrm{tr}}$ and $X_{\mathrm{val}}$, in which $X_{\mathrm{tr}}$ is only used to calculate the updated shared parameters $\theta_{s|i}^{t+1}$. Assuming both tasks have distinct loss functions that do not directly depend on each other during training, we can move the second expectation inside the loss function to obtain the middle term. By using the fact that $X_{\mathrm{tr}}$ and $X_{\mathrm{val}}$ are identically distributed, thus yielding the same expectation, we can move from the middle term to the rightmost term. Hence, the inter-task affinity computed on the training dataset would approximately equal the inter-task affinity computed on the validation set.

**How do inter-task affinities change over the course of training?** Our analysis on CelebA and Taskonomy suggests inter-task affinities change throughout training, but no obvious patterns emerge related to how inter-task affinities change as a function of time across tasks. Nevertheless, our analysis indicates certain tasks exhibit higher-than-average inter-task affinity and tend to maintain this difference throughout training. We visualize this effect in Figure 4 for three tasks in the CelebA dataset. Figure 4(left) shows that task a7 exhibits significantly higher affinity onto task a1 than any other task in the network, and in a similar comparison, Figure 4(right) indicates that tasks {a5, a7} manifest significantly higher affinity onto a6 than any other task. Shifting our focus to a5 in Figure 4(center), we find {a6, a7} display markedly higher affinity, and a1 sometimes displays higher affinity, onto a5 than the other tasks in the network.

One additional observation not captured in Figure 4 relates to the initial steps of training. At this stage of convergence, all tasks seem to manifest positive and similar inter-task affinity. We postulate the representations learned by the model at this early stage may be common among all tasks. For example, the learned representations may extract general-purpose facial characteristics, but specialization occurs quickly thereafter.

**Can changes in hyperparameters affect task groupings?** We define three settings: (i) typical, the base setting, (ii) b=0.5x, or halving the batch size, and (iii) lr=2x, or increasing the learning rate by a factor of 2. To determine the ground-truth task groupings, we train all 511 combinations of task groupings from our 9-task subset of CelebA and select the groups in each setting with the lowest total test error. Our aim is to assess the extent to which task groupings chosen in setting b=0.5x or lr=2x generalize to the typical setting.

Our results are summarized in Table 3. They indicate that simply changing the batch size or learning rate of a model may change which tasks should be trained together, with the groupings found by lr=2x exhibiting worse generalization than those found by b=0.5x. This result suggests that how tasks should be trained together does not simply depend on the relationships among tasks, but also on detailed aspects of the model and training. It is notably difficult to build intuition for the latter, illustrating the need to develop automated methods that can take into account these nuances.

| Budget | Improvement in Test Accuracy Relative to Optimal Groupings | |
| --- | --- | --- |
| | b=0.5x | lr=2x |
| 2-groups | -0.61% | -1.22% |
| 3-groups | -0.25% | -2.90% |
| 4-groups | -1.87% | -3.21% |

Table 3: Accuracy of task groupings found by b=0.5x and lr=2x relative to the typical setting.

## 6 Conclusion

In this work, we present an approach to quantify inter-task affinity in a single training run and show how this quantity can be used to systematically determine which tasks should train together in multi-task networks. Our empirical findings indicate our approach is highly competitive. It outperforms multi-task training augmentations like Uncertainty Weights, GradNorm, and PCGrad, and performs competitively with state-of-the-art task grouping methods like HOA, while improving computational efficiency by over an order of magnitude. Further, our findings are supported by extensive analysis that suggests inter-task affinity scores can find close to optimal auxiliary tasks, and in fact, implicitly measure generalization capability among tasks.

A plethora of research has been undertaken to design better multi-task learning architectures, or improve the optimization dynamics within multi-task learning systems, with relatively little work addressing the question of which tasks should train together in the first place. It is our hope this work renews interest in this domain, and given the sensitivity of task groupings to even small changes in hyperparameters, encourages the development of efficient and automatic methods to identify which tasks should train together in multi-task learning networks.

## 7 Broader Impact

Efficiently identifying task groupings in multi-task learning has the potential to save significant time and computational resources in both academic and industry environments. Despite this benefit, there are several risks associated with this work. In particular, inter-task affinities can be mistakenly interpreted as "task similarity", and incorrectly create an association and/or causation relationship among tasks with high mutual inter-task affinity scores. This association would be especially problematic for datasets involving sensitive prediction quantities related to race, gender, religion, age, status, physical traits, etc., where inter-task affinities could be mistakenly used to support an unfounded conclusion that attempts to posit similarity among tasks. That said, we believe acknowledging these risks mitigates their potential for abuse, and the benefit from this work — most notably decreasing computational resources by over an order of magnitude compared with a state-of-the-art task grouping method while performing competitively in terms of accuracy — merits its dissemination.

## Acknowledgement

We thank Vince Gatto, Bing-Rong Lin, Nick Bridle, Li Wei, Shawn Andrews, Yuan Gao, and Yuyan Wang for their thoughtful discussion related to a precursor of this work. We also thank James Chen and Linda Wang for providing feedback on an earlier draft of this paper. Lastly, we would like to recognize Zirui Wang for technical support with running experiments as well as our anonymous NeurIPS reviewers for their thoughtful feedback and clear desire to improve this work.

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
