# Appendix

## A  Proofs of Theoretical Results

### A.1  Proof of Lemma 1

**Lemma 1.** *Let $L_a$ be a $\alpha$-strongly convex and $\beta$-strongly smooth loss function. Given that task $b$ induces higher inter-task affinity than task $c$ on task $a$, the following inequality holds:*

$$g_a \cdot g_c - \frac{\beta \eta}{2} \|g_c\|^2 + \frac{\alpha \eta}{2} \|g_b\|^2 \leq g_a \cdot g_b \qquad (2)$$

*Proof.* Let $\theta$ be the initial parameters. Let $\theta_b^+ := \theta - \eta \, g_b$ and $\theta_c^+ := \theta - \eta \, g_c$ denote the updated shared parameters using gradient of task $b$ and $c$, respectively. From inter-task affinity, we have

$$\mathcal{Z}_{b \to a} = 1 - \frac{L_a(\theta_b^+)}{L_a(\theta)} \geq 1 - \frac{L_a(\theta_c^+)}{L_a(\theta)} = \mathcal{Z}_{c \to a}$$

Thus, we have

$$L_a(\theta_b^+) \leq L_a(\theta_c^+)$$

From the strong convexity and strong smoothness assumptions, we can respectively lower-bound and upper-bound the first and second terms. Thus, we have

$$L_a(\theta) - \eta \, g_a \cdot g_b + \frac{\alpha \eta^2}{2} \|g_b\|^2 \leq L_a(\theta) - \eta \, g_a \cdot g_c + \frac{L \eta^2}{2} \|g_c\|^2$$

Rearranging the terms yields the result. $\qquad \square$

### A.2  Proof of Proposition 1

**Proposition 1.** *Let $L_a$ be a $\alpha$-strongly convex and $\beta$-strongly smooth loss function. Let $\eta \leq \frac{1}{\beta}$ be the learning rate. Suppose that in a given step, task $b$ has higher inter-task affinity than task $c$ on task $a$. Moreover, suppose that the gradients have equal norm, i.e. $\|g_a\| = \|g_b\| = \|g_c\|$. Then, taking a gradient step on the parameters using the combined gradient of $a$ and $b$ reduces $L_a$ more so than taking a gradient step on the parameters using the combined gradient of $a$ and $c$, given that $\cos(g_a, g_c) \leq \frac{\eta}{4} \frac{\alpha \beta}{\beta - \alpha} - 1$ where $\cos(u, v) := \frac{u \cdot v}{\|u\| \|v\|}$ is the cosine similarity between $u$ and $v$.*

*Proof.* Applying the strong smoothness upper-bound on the updated loss using the combined gradient $g_a + g_b$, we have

$$L_a(\theta - \eta \,(g_a + g_b)) \leq L_a(\theta) - \eta \, g_a \cdot (g_a + g_b) + \frac{\eta^2 \beta}{2} \|g_a + g_b\|^2$$

$$= L_a(\theta) + (\eta^2 \beta - \eta) \, g_a \cdot g_b + (\frac{\eta^2 \beta}{2} - \eta) \|g_a\|^2 + \frac{\eta^2 \beta}{2} \|g_b\|^2$$

We would like to show that the last line is less than or equal to a lower-bound on the loss obtained using the combined gradient $g_a + g_c$

$$L_a(a) - \eta \, g_a \,(a_a + g_c) + \frac{\eta^2 \alpha}{2} \|g_a + g_c\|^2 \leq L_a(\theta - \eta \,(g_a + g_c))$$

Eliminating the common terms, we would like to show the following inequality holds

$$(\eta \beta - 1) \, g_a \cdot g_b + \frac{\eta \beta}{2}(\|g_a\|^2 + \|g_b\|^2) \leq (\eta \alpha - 1) \, g_a \cdot g_c + \frac{\eta \alpha}{2}(\|g_a\|^2 + \|g_c\|^2) \, .$$

Using $\eta \leq \frac{1}{\beta}$, the first term becomes negative. Replacing for $g_a \cdot g_b$ using Eq. (2), it suffices to show the following inequality

$$(\eta \beta - 1) \, (g_a \cdot g_c + \frac{\eta \alpha}{2}(\|g_b\|^2) - \frac{\eta \beta}{2}(\|g_c\|^2)) + \frac{\eta \beta}{2}(\|g_a\|^2 + \|g_b\|^2)$$

$$\leq (\eta \alpha - 1) \, g_a \cdot g_c + \frac{\eta \alpha}{2}(\|g_a\|^2 + \|g_c\|^2) \, .$$

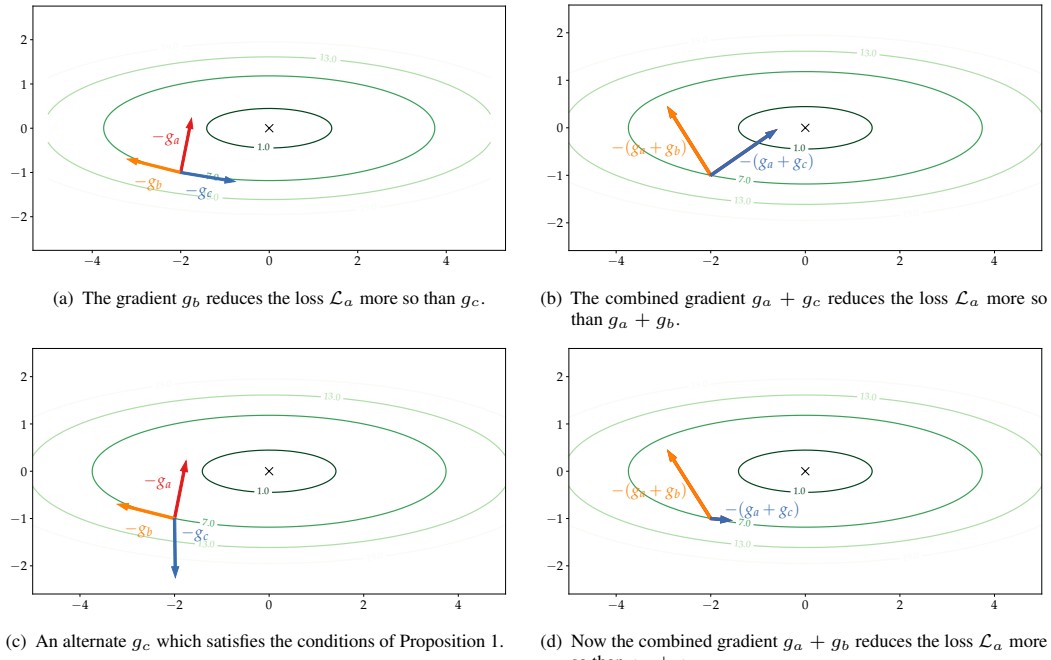

(a) The gradient $g_b$ reduces the loss $\mathcal{L}_a$ more so than $g_c$.

(b) The combined gradient $g_a + g_c$ reduces the loss $\mathcal{L}_a$ more so than $g_a + g_b$.

(c) An alternate $g_c$ which satisfies the conditions of Proposition 1.

(d) Now the combined gradient $g_a + g_b$ reduces the loss $\mathcal{L}_a$ more so than $g_a + g_c$.

Figure 5: A counterexample on a quadratic loss function where the task grouping based on inter-task similarity results in an inferior performance.

Rearranging the terms, it suffices to show

$$0 \leq \|g_a\|^2(\frac{\eta}{2}(\alpha - \beta)) + \|g_b\|^2(-\frac{\eta}{2}\alpha(\eta\beta - 1)) + \|g_c\|^2(\frac{\eta}{2}(\alpha - \beta) + \frac{\eta^2}{2}\beta^2) + \eta(\alpha - \beta)\, g_a \cdot g_c\,.$$

Under the mild assumption that $\|g_a\| = \|g_b\| = \|g_c\|$, we can rearrange the terms to obtain $\cos(g_a, g_c) \leq \frac{\eta}{4}\frac{\beta/\alpha}{\beta/\alpha - 1} - 1$, where $\cos(u, v) := \frac{u \cdot v}{\|u\|\,\|v\|}$ is the cosine similarity between $u$ and $v$. $\quad\square$

### A.3 Quadratic Counterexample

We provide a counterexample in a multi-task setup using a quadratic loss function. The loss function for the task $a$ is defined as $\mathcal{L}_a(x_1, x_2) = \frac{1}{2}(x_1^2 + 10\,x_2^2)$. For this loss function, $\alpha = 1$ and $\beta = 10$. Also, the global minimum of the loss is at $(x_1, x_2) = (0, 0)$. We set the initial point to $(x_1, x_2) = (-2, -1)$ with a loss value of 7. Figure 5(a) shows the level sets of the loss function $\mathcal{L}_a$, along with the negative task gradient $-g_a$. We also plot two additional negative gradients, namely $-g_b$ and $-g_c$, belonging to the tasks $b$ and $c$, respectively. The auxiliary task gradients $g_b$ and $g_c$ are chosen to have the same gradient norm as $g_a$, but pointed along the vectors $[8, -2]$ and $[-12, 2]$, respectively. Using a learning rate of $\eta = 0.09 < \frac{1}{\beta}$, the gradient of task $b$ at this point reduces the value of the loss $\mathcal{L}_a$ more so than the gradient of task $c$. Specifically, the value of the loss after a gradient step using $g_b$ amounts to 6.96 whereas using $g_c$, we obtain a loss value of 6.98. However, this ordering does not hold when combining the gradients, i.e. using the combined gradient $g_a + g_b$ results in a loss value 6.09 of whereas the combined gradient $g_a + g_c$ yields a loss of 6.07 (Figure 5(b)).

Alternatively, in Figure 5(c) we consider a gradient $g_c$ along $[-0.2, 15]$ which also satisfies the second condition of Proposition 1, namely $\cos(a, c) \leq \frac{\eta}{4}\frac{\beta/\alpha}{\beta/\alpha - 1}$ (while $\|g_a\| = \|g_c\|$). For this choice of $g_c$ and as a result of Proposition 1, the fact that $g_b$ reduces the loss $\mathcal{L}_a$ more so than $g_c$ (6.96 vs 7.96) implies $g_a + g_b$ also reduces the loss more so than $g_a + g_c$ (6.09 vs 6.98, see Figure 5(d)).

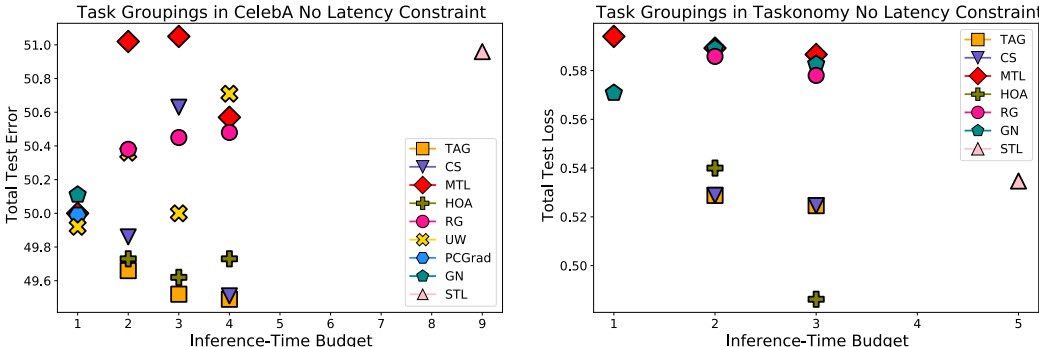

Figure 6: (Left) average classification error for 2, 3, and 4-split task groupings for the subset of 9 tasks in CelebA. (Right) total test loss for 2 and 3-split task groupings for the subset of 5 tasks in Taskonomy. The x-axis is the inference-time memory budget relative to the number of parameters in the baseline MTL model from Section 5.

# B    Additional Experimental Results

We provide additional experimental results to supplement our empirical analysis of TAG in Section 5. In particular, we evaluate task groupings without a fixed inference-time latency constraint, supply extra information related to the CelebA and Taskonomy analyses, and offer additional discussion on the ablation studies presented in Section 5.2.

## B.1    Task Grouping Evaluation Without Latency Constraint

We analyze the effect of removing the inference-time latency constraint from our problem definition. Without this limitation, we can scale up the size of our multi-task learning baselines to equal the total number of parameters used in each task grouping. Similar to [46], we choose to scale capacity by increasing the number of channels in each convolutional layer. A 2-splits task grouping would then correspond with a multi-task model with 2 times the number of channels in each conv layer. On the CelebA dataset, running PCGrad with double the number of channels resulted in an out-of-memory (OOM) error on a 16 GB TeslaV100 GPU. When implemented with distributed training, we received a runtime error. As a result, we do not include PCGrad results in this analysis, and this particular method would also likely surpass typical computational budget constraints due to its high memory usage.

Our results are summarized in Figure 6. Similar to the findings presented in Figure 3 of Section 5, the task grouping approaches continue to outperform simply training all tasks together, as well as optimization augmentations like Uncertainty Weights and GradNorm. For CelebA, increasing the number of channels in the layers of our ResNet model actually reduces performance, indicating our model is already at near-optimal capacity for this dataset. This is reasonable given our tuning of the CelebA model architecture and hyperparameters to maximize MTL performance. For Taskonomy, and similar to the results presented in [46], we find scaling multi-task model capacity does not meaningfully improve MTL or GradNorm performance.

## B.2    Additional CelebA Task Grouping Results

In this section, we provide further detail into our experimental results for the CelebA dataset. We present the raw values used to create Figure 3 (left) and Figure 6 (left) as well as underpin the Section 5.2 ablation studies in Table 4, Table 5, Table 6, Table 7, and Table 8. All quantities are averaged across three independent runs, and we report mean and standard error. A task within a group is highlighted in bold when this task is chosen to "serve" from this assigned task grouping. Duplicate tasks that are not bolded are only used to assist in the training other tasks.

| CelebA Baseline Methods | | | | | | | | | | |
|---|---|---|---|---|---|---|---|---|---|---|
| Method | a1 | a2 | a3 | a4 | a5 | a6 | a7 | a8 | a9 | Total Error |
| MTL | 6.54 ± 0.026 | 11.09 ± 0.009 | 4.19 ± 0.017 | 12.59 ± 0.085 | 2.60 ± 0.0.003 | 2.73 ± 0.128 | 4.81 ± 0.010 | 4.74 ± 0.012 | 0.70 ± 0.007 | 50.00 |
| STL | 6.56 ± 0.003 | 11.37 ± 0.009 | 4.19 ± 0.025 | 12.58 ± 0.102 | 2.69 ± 0.017 | 3.06 ± 0.010 | 4.97 ± 0.006 | 4.83 ± 0.010 | 0.71 ± 0.007 | 49.99 |
| UW [28] | 6.51 ± 0.038 | 11.43 ± 0.034 | 4.18 ± 0.015 | 11.91 ± 0.132 | 2.50 ± 0.028 | 2.95 ± 0.010 | 4.81 ± 0.026 | 4.89 ± 0.028 | 0.74 ± 0.007 | 49.92 |
| GradNorm [10] | 6.44 ± 0.033 | 11.09 ± 0.0021 | 4.01 ± 0.051 | 12.38 ± 0.097 | 2.65 ± 0.022 | 2.96 ± 0.017 | 4.89 ± 0.015 | 4.87 ± 0.003 | 0.81 ± 0.009 | 50.11 |
| PCGrad [53] | 6.57 ± 0.015 | 10.95 ± 0.020 | 4.04 ± 0.015 | 12.73 ± 0.033 | 2.67 ± 0.013 | 2.87 ± 0.021 | 4.76 ± 0.006 | 4.76 ± 0.015 | 0.64 ± 0.010 | 49.99 |

Table 4: Mean and standard error for benchmark methods run on CelebA.

| Inference Time Budget = 2 Splits Task Groupings | | | | | | | | | | | |
|---|---|---|---|---|---|---|---|---|---|---|---|
| Method | Splits | a1 | a2 | a3 | a4 | a5 | a6 | a7 | a8 | a9 | Total Error |
| CS | group 1 | — | — | — | — | 2.60 ± 0.010 | 3.05 ± 0.007 | — | — | — | 49.86 |
| | group 2 | 6.55 ± 0.009 | 11.19 ± 0.020 | 4.10 ± 0.012 | 12.02 ± 0.029 | 2.57 ± 0.007 | — | 4.78 ± 0.015 | 4.85 ± 0.006 | 0.73 ± 0.010 | |
| HOA | group 1 | — | — | — | — | 2.59 ± 0.006 | — | 4.71 ± 0.000 | — | — | 49.73 |
| | group 2 | 6.49 ± 0.037 | 11.34 ± 0.022 | 4.25 ± 0.052 | 11.76 ± 0.090 | — | 3.00 ± 0.022 | — | 4.91 ± 0.059 | 0.69 ± 0.009 | |
| TAG | group 1 | 6.39 ± 0.006 | — | — | — | — | — | 4.79 ± 0.006 | — | — | 49.66 |
| | group 2 | — | 11.10 ± 0.065 | 4.16 ± 0.003 | 12.29 ± 0.202 | 2.55 ± 0.025 | 2.94 ± 0.015 | — | 4.69 ± 0.026 | 0.74 ± 0.013 | |
| Optimal | group 1 | 6.60 ± 0.009 | 11.21 ± 0.017 | 4.40 ± 0.007 | 11.91 ± 0.051 | 2.60 ± 0.009 | 2.87 ± 0.003 | 4.81 ± 0.015 | 4.58 ± 0.009 | — | 49.37 |
| | group 2 | — | 11.14 ± 0.044 | 4.03 ± 0.012 | 11.90 ± 0.020 | — | — | — | — | 0.75 ± 0.018 | |

Table 5: Two-split task groupings in CelebA. We report mean and standard error.

| Inference Time Budget = 3 Splits Task Groupings | | | | | | | | | | | |
|---|---|---|---|---|---|---|---|---|---|---|---|
| Method | Splits | a1 | a2 | a3 | a4 | a5 | a6 | a7 | a8 | a9 | Total Error |
| CS | group 1 | — | — | — | — | 2.60 ± 0.010 | 3.05 ± 0.007 | — | — | — | 50.63 |
| | group 2 | 6.39 ± 0.006 | — | — | — | — | — | 4.79 ± 0.006 | — | — | |
| | group 3 | — | 11.20 ± 0.031 | 4.09 ± 0.010 | 12.97 ± 0.015 | — | — | — | 4.82 ± 0.018 | 0.73 ± 0.003 | |
| HOA | group 1 | — | — | — | — | 2.59 ± 0.006 | — | 4.71 ± 0.000 | — | — | 49.73 |
| | group 2 | — | 11.08 ± 0.017 | — | 12.20 ± 0.067 | — | — | — | — | — | |
| | group 3 | 6.55 ± 0.012 | — | 4.15 ± 0.013 | — | 2.70 ± 0.012 | 2.84 ± 0.003 | 4.90 ± 0.015 | 4.76 ± 0.015 | 0.75 ± 0.013 | |
| TAG | group 1 | 6.39 ± 0.006 | — | — | — | — | — | 4.79 ± 0.006 | — | — | 49.52 |
| | group 2 | — | 11.08 ± 0.017 | — | 12.20 ± 0.067 | — | — | — | — | — | |
| | group 3 | — | 11.11 ± 0.127 | 4.08 ± 0.006 | — | 2.52 ± 0.025 | 2.96 ± 0.050 | 4.98 ± 0.058 | 4.73 ± 0.015 | 0.78 ± 0.009 | |
| Optimal | group 1 | 6.34 ± 0.067 | — | — | — | 2.57 ± 0.012 | 2.91 ± 0.019 | 4.74 ± 0.031 | 4.70 ± 0.017 | 0.78 ± 0.012 | 48.92 |
| | group 2 | — | 11.14 ± 0.044 | 4.03 ± 0.012 | 11.90 ± 0.020 | — | — | — | — | 0.75 ± 0.018 | |
| | group 3 | 6.25 ± 0.045 | — | — | — | — | — | 4.67 ± 0.009 | — | 0.71 ± 0.006 | |

Table 6: Three-split task groupings in CelebA. We report mean and standard error.

| Inference Time Budget = 4 Splits Task Groupings | | | | | | | | | | | |
|---|---|---|---|---|---|---|---|---|---|---|---|
| Method | Splits | a1 | a2 | a3 | a4 | a5 | a6 | a7 | a8 | a9 | Total Error |
| CS | group 1 | — | — | — | — | 2.60 ± 0.010 | 3.05 ± 0.007 | — | — | — | 49.51 |
| | group 2 | 6.39 ± 0.006 | — | — | — | — | — | 4.79 ± 0.006 | — | — | |
| | group 3 | — | 11.08 ± 0.017 | — | 12.20 ± 0.067 | — | — | — | — | — | |
| | group 4 | — | 11.00 ± 0.013 | 4.07 ± 0.012 | 12.40 ± 0.045 | — | — | 4.96 ± 0.031 | 4.62 ± 0.009 | 0.72 ± 0.007 | |
| HOA | group 1 | — | 11.08 ± 0.017 | — | 12.20 ± 0.067 | 2.59 ± 0.006 | — | 4.71 ± 0.000 | — | — | 49.73 |
| | group 2 | — | — | — | — | 2.70 ± 0.006 | — | — | — | — | |
| | group 3 | 6.63 ± 0.024 | — | — | — | 2.58 ± 0.023 | 2.88 ± 0.045 | 4.90 ± 0.065 | 4.74 ± 0.026 | 0.76 ± 0.012 | |
| | group 4 | — | 11.25 ± 0.049 | 4.15 ± 0.024 | — | — | — | — | — | — | |
| TAG | group 1 | 6.39 ± 0.006 | — | — | — | — | — | 4.79 ± 0.006 | — | — | 49.49 |
| | group 2 | — | 11.08 ± 0.017 | — | 12.20 ± 0.067 | — | — | — | — | — | |
| | group 3 | — | — | — | — | 2.63 ± 0.003 | 2.99 ± 0.015 | 4.75 ± 0.012 | — | — | |
| | group 4 | — | 11.00 ± 0.013 | 4.07 ± 0.012 | 12.40 ± 0.045 | — | — | 4.96 ± 0.031 | 4.62 ± 0.009 | 0.72 ± 0.007 | |
| Optimal | group 1 | — | 11.20 ± 0.049 | 4.09 ± 0.022 | 11.77 ± 0.197 | — | 2.90 ± 0.020 | 4.85 ± 0.041 | — | 0.75 ± 0.010 | 48.57 |
| | group 2 | — | — | 4.00 ± 0.023 | — | — | — | 4.67 ± 0.009 | — | 0.71 ± 0.006 | |
| | group 3 | 6.25 ± 0.045 | — | — | — | — | — | — | — | — | |
| | group 4 | — | 10.96 ± 0.045 | — | 12.74 ± 0.136 | 2.56 ± 0.031 | — | — | 4.75 ± 0.072 | 0.77 ± 0.022 | |

Table 7: Four-split task groupings in CelebA. We report mean and standard error.

| CelebA High Capacity Performance | | | | | | | | | | | |
|---|---|---|---|---|---|---|---|---|---|---|---|
| Method | Capacity | a1 | a2 | a3 | a4 | a5 | a6 | a7 | a8 | a9 | Total Error |
| MTL | 2x | 6.42 ± 0.007 | 10.76 ± 0.070 | 4.35 ± 0.010 | 13.01 ± 0.152 | 2.66 ± 0.015 | 3.05 ± 0.009 | 4.76 ± 0.038 | 5.20 ± 0.047 | 0.80 ± 0.007 | 51.02 |
| | 3x | 6.43 ± 0.032 | 11.65 ± 0.128 | 4.21 ± 0.060 | 12.69 ± 0.918 | 2.61 ± 0.029 | 2.94 ± 0.009 | 4.87 ± 0.067 | 4.86 ± 0.064 | 0.80 ± 0.034 | 51.05 |
| | 4x | 6.91 ± 0.055 | 11.23 ± 0.060 | 4.31 ± 0.032 | 12.51 ± 0.104 | 2.57 ± 0.021 | 3.01 ± 0.009 | 4.59 ± 0.036 | 4.81 ± 0.018 | 0.64 ± 0.012 | 50.57 |
| UW | 2x | 6.40 ± 0.006 | 11.01 ± 0.023 | 4.36 ± 0.015 | 12.54 ± 0.022 | 2.63 ± 0.0.013 | 3.03 ± 0.003 | 4.68 ± 0.007 | 5.00 ± 0.015 | 0.71 ± 0.012 | 50.36 |
| | 3x | 6.36 ± 0.050 | 11.32 ± 0.063 | 4.27 ± 0.020 | 12.12 ± 0.019 | 2.51 ± 0.012 | 2.94 ± 0.037 | 4.93 ± 0.045 | 4.76 ± 0.037 | 0.79 ± 0.006 | 50.00 |
| | 4x | 6.70 ± 0.146 | 11.49 ± 0.107 | 4.37 ± 0.009 | 12.35 ± 0.120 | 2.63 ± 0.021 | 3.07 ± 0.057 | 4.61 ± 0.023 | 4.83 ± 0.013 | 0.64 ± 0.018 | 50.71 |
| GN | 2x | 6.38 ± 0.038 | 10.98 ± 0.031 | 4.13 ± 0.029 | 12.57 ± 0.095 | 2.71 ± 0.0.000 | 3.00 ± 0.043 | 4.73 ± 0.009 | 5.21 ± 0.146 | 0.86 ± 0.019 | 50.57 |
| | 3x | 6.38 ± 0.021 | 11.67 ± 0.217 | 4.35 ± 0.075 | 12.35 ± 0.248 | 2.61 ± 0.020 | 3.00 ± 0.089 | 4.93 ± 0.056 | 4.86 ± 0.006 | 0.85 ± 0.037 | 51.00 |
| | 4x | 6.56 ± 0.063 | 11.49 ± 0.261 | 4.33 ± 0.030 | 13.29 ± 0.646 | 2.57 ± 0.052 | 3.07 ± 0.067 | 4.82 ± 0.087 | 4.82 ± 0.057 | 0.78 ± 0.003 | 51.74 |

Table 8: Mean and standard error for CelebA high capacity experiments.

## B.3 Additional Taskonomy Task Grouping Results

We also report the raw scores used in our empirical analysis of Taskonomy to create Figure 3 (right) and Figure 6 (right) in Table 9, Table 10, Table 11, and Table 12. Given the size of the Taskonomy dataset, and computational cost associated with running a single model (approximately 146 Tesla V100 GPU hours for the MTL baseline), we evaluate only a single run.

| Baseline Methods on Taskonomy | | | | | | |
|---|---|---|---|---|---|---|
| Method | s | d | n | t | k | Total Test Loss |
| MTL | 0.0586 | 0.2879 | 0.1076 | 0.0428 | 0.1079 | 0.5940 |
| STL | 0.0509 | 0.2616 | 0.0975 | 0.0337 | 0.0910 | 0.5347 |
| GN | 0.0542 | 0.2818 | 0.1011 | 0.0305 | 0.1032 | 0.5708 |

Table 9: Baseline methods in Taskonomy.

| Inference Time Budget = 2 Splits Task Groupings | | | | | | | |
|---|---|---|---|---|---|---|---|
| Method | Splits | s | d | n | t | k | Total Test Loss |
| CS | group 1 | **0.532** | **0.2527** | **0.1064** | — | — | 0.5288 |
| | group 2 | — | — | — | **0.0232** | **0.0933** | |
| HOA | group 1 | **0.0603** | **0.2725** | **0.1075** | **0.0429** | — | 0.5400 |
| | group 2 | — | — | 0.1110 | — | **0.0568** | |
| TAG | group 1 | **0.532** | **0.2527** | **0.1064** | — | — | 0.5288 |
| | group 2 | — | — | — | **0.0232** | **0.0933** | |
| Optimal | group 1 | **0.0532** | **0.2527** | **0.1064** | — | — | 0.5176 |
| | group 2 | — | — | 0.1096 | **0.0271** | **0.0750** | |

Table 10: Two-split task groupings in Taskonomy.

| Inference Time Budget = 3 Splits Task Groupings | | | | | | | |
|---|---|---|---|---|---|---|---|
| Method | Splits | s | d | n | t | k | Total Test Loss |
| CS | group 1 | **0.528** | 0.2636 | — | — | — | 0.5246 |
| | group 2 | — | — | — | **0.0232** | **0.0933** | |
| | group 3 | — | **0.2551** | 0.1002 | — | — | |
| HOA | group 1 | **0.0532** | **0.2527** | **0.1064** | — | — | 0.4923 |
| | group 2 | — | — | 0.1110 | — | **0.0568** | |
| | group 3 | — | — | — | **0.0232** | 0.0933 | |
| TAG | group 1 | **0.528** | 0.2636 | — | — | — | 0.5246 |
| | group 2 | — | — | — | **0.0232** | **0.0933** | |
| | group 3 | — | **0.2551** | 0.1002 | — | — | |
| Optimal | group 1 | 0.0532 | **0.2527** | 0.1064 | — | — | 0.4862 |
| | group 2 | **0.0500** | — | **0.1025** | **0.0242** | — | |
| | group 3 | — | — | 0.1110 | — | **0.0568** | |

Table 11: Three-split task groupings in Taskonomy.

| Taskonomy High Capacity Performance | | | | | | |
|---|---|---|---|---|---|---|
| Method | s | d | n | t | k | Total Test Loss |
| MTL (2x) | 0.00536 | 0.2664 | 0.1078 | 0.0441 | 0.11.73 | 0.5892 |
| MTL (3x) | 0.0538 | 0.2891 | 0.1090 | 0.0363 | 0.0984 | 0.5866 |
| GN (2x) | 0.0552 | 0.2977 | 0.1013 | 0.0289 | 0.1060 | 0.5891 |
| GN (3x) | 0.0570 | 0.2806 | 0.1044 | 0.0401 | 0.1006 | 0.5827 |

Table 12: Performance of high capacity models on Taskonomy.

## B.4   Supplementary Information on Ablation Studies

In this section, we expand on our analysis into whether inter-task affinity correlates with optimal task groupings. Notably, we offer additional insight into the failure case of TAG as well as a limitation that arises from computing the change in loss on the training dataset rather than the validation dataset. Finally, we provide an additional dimension of analysis into the robustness of inter-task affinity measurements. In particular, *is the inter-task affinity between two tasks maintained even in the presence of a different set of tasks?*

**Are inter-task affinities maintained across different sets of tasks?** For example, if one model trains with tasks {A, B, C, D} and another with tasks {B, C, E, F}, would the inter-task affinity between B and C be similar in both cases? Our empirical findings indicate inter-task affinity scores

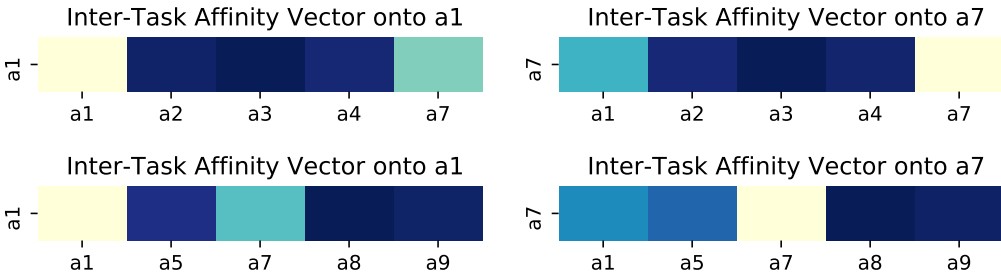

Figure 7: (Left) inter-task affinity onto task a1 when trained with (top) {a1, a2, a3, a4, a7} and (bottom) {a1, a5, a7, a8, a9}. Note, the inter-task affinity of a7 onto a1 is maintained across both groups. (Right) inter-task affinity onto task a7 when trained with (top) {a1, a2, a3, a4, a7} and (bottom) {a1, a5, a7, a8, a9}. Note, the inter-task affinity of a1 onto a7 is maintained across both groups. Lighter coloring signify higher inter-task affinities.

between two tasks are largely maintained even in the presence of entirely different task sets on the CelebA dataset. We visualize this effect for two such cases in Figure 7. This analysis presented in Figure 7(left) indicates that the inter-task affinity from a7 onto a1 is consistent across two different sets of tasks. Similarly, the inter-task affinity from a1 onto a7 is largely preserved as shown in Figure 7(right).

However, there is no guarantee our empirical observations will extend to all cases. The second task set may contain a task that significantly decreases model performance, or even causes divergence. In this event, it is likely that the inter-task affinity exhibited between two tasks in the first set will significantly differ from the second set.

**Does our measure of inter-task affinity correlate with optimal task groupings?** Expanding on our analysis in Section 5.2, we present the pairwise training-level inter-task affinity scores for the CelebA and Taskonomy datasets in Figure 8. Our analysis indicates three "groups" naturally appear in CelebA composed of {a1, a7}, {a2,a3,a4}, and {a5, a6, a7}. Meanwhile on Taskonomy, two "groups" composed of {segmentation, depth, normals} and {edges, keypoints} naturally form. The Network Selection Algorithm operates on this information to assign tasks to networks.

Expanding our analysis into the failure case of TAG to select the best auxiliary task for a8 as presented in Table 1, Figure 8 indicates no other task in our CelebA task set exhibits especially high (or low) inter-task affinity onto a8. We further analyze this characteristic in Table 13 which presents the difference in normalized inter-task

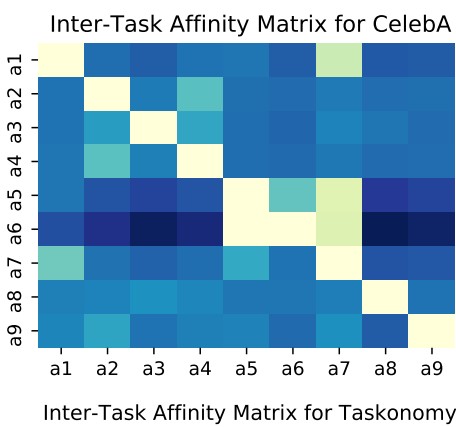

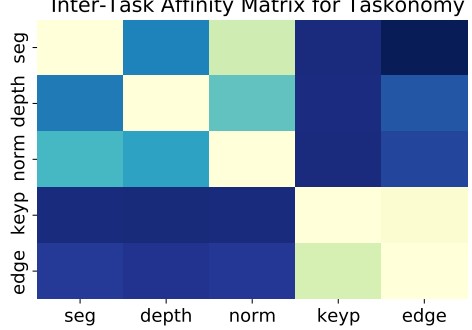

Figure 8: (Top) inter-task affinity on CelebA. (Bottom) inter-task affinity on Taskonomy. Lighter colors signify higher inter-task affinities.

affinity values for each attribute in our CelebA analysis. Note the affinity onto a8 is significantly less than the affinity onto any other task and 4 times smaller than the next smallest difference. We posit the reason TAG fails to select a positive group for a8 is due to the fact that no other task significantly decreases (or increases) the loss of a8 throughout the course of training. Tasks which exhibit similar characteristics are also likely to be difficult cases for TAG when finding a task's best training partner.

**Is change in train loss comparable to change in validation loss?**
From Section 5.2, we show $\mathcal{Z}_{i \to j}^t$ computed on the training set is approximately equal to $\mathcal{Z}_{i \to j}^t$ computed on the validation set when $i \neq j$:

$$\mathbb{E}_{\text{val}}[\mathbb{E}_{\text{tr}}[\mathcal{L}_j(X^t_{\text{val}}, \theta^{t+1}_{s|i}, \theta^t_j)]] \approx \mathbb{E}_{\text{val}}[\mathcal{L}_j(X^t_{\text{val}}, \mathbb{E}_{\text{tr}}[\theta^{t+1}_{s|i}], \theta^t_j)] = \mathbb{E}_{\text{tr}}[\mathcal{L}_j(X^t_{\text{tr}}, \mathbb{E}_{\text{tr}}[\theta^{t+1}_{s|i}], \theta^t_j)] \,.$$

A corollary to this result is the above approximation does not hold when $j = i$: comparing task $i$'s capacity to decrease it's own train loss is not comparable with task $j$'s capacity to decrease the train loss of task $i$. As a result, TAG acting on the training dataset will never group a task by itself. While we find a single-task group is never optimal in any of our experiments, one can tradeoff a small decrease in efficiency from loading the validation dataset during training with the capacity of TAG to select single-task groupings.

| Task | max - min |
|------|-----------|
| a1 | 0.35 |
| a2 | 0.39 |
| a3 | 0.16 |
| a4 | 0.31 |
| a5 | 0.47 |
| a6 | 0.31 |
| a7 | 0.43 |
| a8 | **0.04** |
| a9 | 0.17 |

Table 13: Difference in normalized inter-task affinity onto each task. Notice the affinity onto $a8$ is **significantly** less than the affinity onto any other task.

## B.5 Experimental Design

In this section, we aim to accurately and precisely describe our experimental design to facilitate reproducibility. We also release our code at github.com/google-research/google-research/tree/master/tag to supplement this written explanation.

## B.6 CelebA

We accessed the CelebA dataset publicly available on TensorFlow datasets under an Apache 2.0 license at https://tensorflow.org/datasets/catalog/celeb_a and filtered the 40 annotated attributes down to a set of 9 attributes for our analysis. Our experiments were run on a combination of Keras [12] and TensorFlow [1].

The encoder architecture is based loosely on ResNet 18 [20] with task-specific decoders being composed of a single projection layer. A coarse architecture search revealed adding additional layers to the encoder and decoder did not meaningfully improve model performance. A learning rate of 0.0005 is used for 100 epochs, with the learning rate being halved every 15 epochs. The learning rate was tuned on the validation split of the CelebA dataset over the set of $\{0.00005, 0.0001, 0.0005, 0.001, 0.005, 0.01\}$. GradNorm [10] alpha was determined by searching over the set of $\{0.01, 0.1, 0.5, 1.0, 1.5, 2.0, 3.0, 5.0\}$, and choosing the alpha with the highest total accuracy on the validation set.

We train until the validation increases for 10 consecutive epochs, load the parameters from the best validation checkpoint, and evaluate on the test set. We use the splits default to TensorFlow datasets of (162,770, 19,867, 19,962) for (Train, Valid, Test). As each model trains for a varying number of epochs, we report the worst-case runtime in Figure 3(left) which approximates the time required to train each method when the model is trained for the full 100 epochs. This is similar to the method in [46] which trains to completion, loads the best weights, and then evaluates on the test set. We choose 100 epochs as our setup mirrors that of [45] on CelebA with the exception of adding an early stopping condition.

## B.7 Taskonomy

Our experiments mirror the settings and hyperparameters of "Setting 2" in [46] by directly implementing TAG and its approximation in the framework provided by the author's official code release (https://github.com/tstandley/taskgrouping at hash dc6c89c269021597d222860406fa0fb81b02a231). The encoder is a modified Xception Network and each task-specific decoder consists of four transposed convolutional layers and four convolutional layers. Further information regarding network specifications and training details can be found in [46].

To mitigate computational requirements and increase accessibility, we replace the 12 TB full+ Taskonomy split used by [46] with an augmented version of the medium Tasknomy split by filtering out buildings with corrupted images and adding additional buildings to replace the corrupted ones. We download the Taskonomy dataset from the official repository (https://github.com/StanfordVL/taskonomy) created by [54] which is released under an MIT license. The final size of our medium+ taskonomy

split is approximately 2.4 TB. The list of buildings used in our analysis is encapsulated within our released code. We reuse the implementation of GradNorm in [46] and follow their settings of $\alpha = 1.5$.

While [46] load the parameters with the lowest validation loss into the model before evaluating on the test set, their early-stopping window size is equal to the total number of epochs. Therefore, each model trains for a full 100 epochs, irrespective of whether the lowest validation loss occurred early or late into training. Accordingly, the runtimes in Figure 3 (right) is the time taken by each cloud instance to completely train the model to 100 epochs.