# OpenReview forum: "Efficiently Identifying Task Groupings for Multi-Task Learning"
_NeurIPS.cc/2021/Conference — NeurIPS 2021 Spotlight_

### Official Review · Reviewer_LXec · 2021-07-12

**Rating:** 6
**Confidence:** 4

**Summary:**

This paper presents a task grouping approach for multi-task learning performed in a single training step. It measures an inter-task affinity by quantifying the effect of a task’s gradient on other tasks. This work also shows a theoretical analysis of the task grouping by the inter-task affinity under some conditions. Experimental results show that the presented approach gives competitive results compared to other multi-task learning counterparts using two benchmark datasets.

**Limitations And Societal Impact:**

The idea of measuring inter-task affinity is simple and the technical novelty is limited. The pairwise affinity is measured by comparing the losses before and after applying the gradient update in the shared body of the network. Since it computes all the pairwise task affinities, I think it takes heavy complexity.

It is unclear and not well described how to split tasks into several groups using the affinity. Based on the score between L152 and L153, how to group tasks, and in L170, how to set k in the method?

The theoretical analysis seems not practical because it assumes a convex setting with a smooth loss function. If using popular deep neural networks, it is questionable the analysis can still be valid.

In L242, how the 9 attributes were selected out of 40 attributes in CelebA?

In Figure 2, I wonder how to collect the results (error, runtime) of the compared methods and whether they are collected under the same experimental setup.

In Table 2, it is unclear what the value means. An exact measure for the relative performance should be provided to reproduce.

In Table 3, it is insufficient to show another set of hyperparameters. A control experiment with respect to multiple different values of hyperparameters (batch size, learning rate) should be provided to show the stability of the proposal.

--

Update:
I have read the authors' responses and other reviewers' comments.
Many of my concerns have been addressed, but reducing the expensive cost for computing pairwise affinity is rather heuristic which is performed by a simple trick, not from the algorithm itself. It would be better to address this point to make the paper stronger.
By considering other reviewers' comments and their views on the work, I agree with them, and my comment is not much deviated from others' opinions. The proposed method can be helpful for practitioners due to its simplicity and promising results.
So, I raise my rating and wouldn't mind if the paper gets accepted.

**Main Review:**

This work focuses on suggesting an inter-class affinity measure rather than proposing other strategies for multi-task learning such as analyzing dataset characteristics, model architectures, etc. Since this work assumes learning tasks in a single architecture (hard-parameter sharing), identifying similar tasks and grouping them are important tasks to improve learning efficiency. It presents a simple approach to identify task groups based on the response of gradient updates in the shared parameters. This strategy leads to comparable performance for two multi-task learning scenarios.

**Time Spent Reviewing:**

3-4

---

> ### Author Response · Authors · 2021-08-08
> **Response to Reviewer LXec**
>
> Thank you for your review! We respond to the points you raise below:
>
> 1. **Computing pairwise task affinity is expensive:** Your intuition is correct -- computing pairwise affinity scores at every step entails significant computational complexity. We investigate this dimension in ablation study **Should Inter Task Affinities Be Computed At Every Step** (Section 5.2) and find there to be significant redundancy in affinity scores across subsequent training steps. Accordingly, we sample inter-task affinities every 10 steps of training which results in a relative speedup of 319% when compared to the runtime of computing pairwise task affinity scores at every step. As a result, the time to completely train a multi-task learning model while computing inter-task affinity increases by 30% when evaluated on both CelebA and Taskonomy.
> 2. **How to select k?** Selecting k is a decision made by a researcher or practitioner, and is a function of the available inference-time parameter budget. Inference-time memory is often fixed and unmovable, and under realistic settings, we anticipate most researchers and practitioners will set k=2 or k=3. Empirically, our results indicate significant improvement from k=2, with diminishing returns as k increases to k=3 or k=4. We will add this discussion to the final version.
> 3. **Network selection algorithm ambiguity:**
> We will reformulate our explanation of the network selection algorithm presented in Section 4.2 to make it more clear. Tangibly, we will include an algorithm box clearly describing the steps. For the purposes of this response, the network selection algorithm selects task groupings to maximize the sum of the inter-task affinity onto each task. The high level steps can be broken down into: (I) for all groups, compute the inter-task affinity onto each task (see L164-168), (II) given a fixed k, select k networks such that each task is present in at least one network, and the sum of the inter-task affinity onto each task is maximized. This problem can be reformulated as a Set Cover (NP-hard) and solved efficiently with a branch-and-bound-like algorithm or by encoding the problem as a binary integer program. To facilitate reproducibility, the algorithm was released in our submitted code, and we will open source the files following conference decisions.
> 4. **Theoretical analysis not practical:** We agree analysis in the convex setting often does not apply to deep learning; however, it may be useful for building theoretical insight. Moreover, we would posit a negative theoretical result, i.e. divergence, or selecting poor task groupings, in the convex setting, would make a method less compelling. Lastly, analysis in the convex setting is not abnormal in multi-task learning research, and is done in prominent (and recent) works such as [1] and [2]. It is also important to the deep learning community at large and has served as the theoretical foundation upon which optimization methods such as Adagrad [3] and Adam [4] were developed.
> 5. **How attributes were selected in CelebA:** We initially selected 9 attributes randomly from CelebA with strong empirical results. However, some of the associations found by our task grouping algorithm could be easily misconstrued (see Section 7 for our reasoning). For instance, the task "age" exhibited very high inter-task affinity with "attractive". Accordingly, we hand picked 9 tasks which we believed were not related to sensitive characteristics. No other analysis was done before selecting the 9 tasks, and once selected, no task was excluded. The default parameters in our released code show the chosen tasks.
> 6. **Experimental setup of Figure 2:** You are correct; all results are collected under the same experimental setup, and we detail the experimental design in Section B.5 of the Appendix. To summarize, we tune the parameters for the multi-task model training all tasks together, and reuse these hyperparameters for all optimization method and task groupings experiments. The experimental results on CelebA were conducted on a single Tesla V100 GPU cloud instance while the experimental results for Taskonomy were conducted on a cloud instance of Tesla V100s. Each model used early stopping on the validation set, and test loss is reported. Runtime is reported as wall-clock time per V100 used, which is comparable across methods as each experiment for a given dataset was run on the same hardware, cloud provider, and used the same architecture.
> 7. **Table 2 relative performance:** We agree this point should be clarified. The expected performance of the random grouping is provided under varying inference-time splits in Table 5, 6, and 7 of the Appendix, and one can work backwards from the percent improvement in total accuracy from Table 2 to determine these values, but we can break out the per-split accuracy for each ablation. We will add a table containing these values in the final version.
> 8. **Table 3 stability:** We believe you may have misinterpreted this section. TAG itself does not have any hyperparameters. Table 3 instead presents empirical findings to motivate the utility of an efficient method for automatically determining task groupings by showing that the best task grouping depends on small changes to model hyperparameters. Because TAG measures task affinities under a particular hyperparameter setting, it can adapt to these differences and thus offer improvement over methods that consider a hyperparameter-agnostic task grouping strategy. We will clarify the purpose of this section in the text to remove any ambiguity.
>
>
> [1] Wang et al. Gradient Vaccine: Investigating and improving multi-task optimization in Massively Multilingual Models, ICLR 2021 (spotlight)
>
> [2] Yu et al. Gradient Surgery for Multi-Task Learning, NeurIPS 2020
>
> [3] Duchi et al. Adaptive Subgradient Methods for Online Learning and Stochastic Optimization, JMLR 2011
>
> [4] Kingma et al. Adam: A Method for Stochastic Optimization, ICLR 2015

---

### Official Review · Reviewer_1wZL · 2021-07-15

**Rating:** 7
**Confidence:** 4

**Summary:**

Given a large set of tasks, the goal of this work is to identify subsets of these tasks that can be trained together to improve performance and avoid harmful interference. In contrast to multi-task research that focuses on building a model that handles _all_ tasks as well as possible, this work looks at the cues acquired from training such a model to quickly assess which tasks to group together. This is done by seeing whether a task's gradient update helps or hurts the losses of other tasks.

**Limitations And Societal Impact:**

I found the "Broader Impact" statement solid, much better than what I'm used to coming across in submissions

**Main Review:**

Main thoughts:

- (+) the proposed strategy for measuring task affinity is well justified and straightforward - would be easy enough to implement and replicate. From a first impression it is unclear that it would necessarily correlate with subsequent multi-task performance, but the experiments back up its utility as a grouping signal
- (+) the case made for efficiency (that this method only requires a single training run of a model with all tasks) is also a strong advantage relative to the cost of other ways to find task groupings
- (+) clear and well written work with thoughtful analysis throughout paper and in ablations
- (-) I do not find 9 attributes on CelebA the most compelling multi-task benchmark (though I understand CelebA is a go-to setting in the multi-task literature). The difference in test error is marginal across these methods and facial attribute classification is such a restricted setting. I think it would be better to see results on diverse multi-domain multi-task settings like Visual Decathlon or VTAB. There is extra overhead in using the proposed method in those settings since all task losses cannot be evaluated in a single inference pass, but seems as though it would still be useful.

I lean positive for this paper, I think that the authors propose a nice, simple strategy for grouping tasks in multi-task training and provide a solid set of experiments and analysis to justify its use.

Additional comments:

- There's an ablation looking at task affinity at different stages of training, the dynamics here seem like they'd be interesting to have a clearer picture of. How does affinity typically change over the course of training, what does it look like for the tasks that match up best, are there different patterns for different pairs of tasks (moving from high -> low affinity or vice versa)
- Are task affinities robust to the presence of other sets of tasks. Say I trained a model with Tasks A, B, C, D and a model with Tasks B, C, E, F. Would the affinity between B + C be similar in both cases?
- Is it possible to see the distribution in performance of random groupings? Were the optimal and worst values derived from an exhaustive search or were they the best/worst to appear in the random sampling?

Typo L224 "hyperparamters", L263 "Segmentic Segmentation"


-----
**Update:** After having read the other reviews and author responses I maintain my original recommendation to accept. I really appreciate the effort in the response along with the attached figures and details that addressed my questions.

**Time Spent Reviewing:**

3

---

> ### Author Response · Authors · 2021-08-08
> **Response to Reviewer 1wZL**
>
> Thank you for your review! We appreciate you finding two typos and offering a kind remark on our Broader Impact statement, especially given our sedulous focus on writing this paragraph. We provide a response to your comments below:
>
> 1. **Inter-task affinity as a function of time:** This is an insightful question. Inter-task affinity does change over the course of training, though tasks which exhibit higher-than-normal inter-task affinity tend to maintain this difference throughout training. We visualize this effect at https://sites.google.com/view/anon-response-to-reviewer-1wzl (website analytics turned off to maintain the anonymity of the reviewer). One relevant observation not reflected in the website figures is that during the initial steps of training (approximately the first half of the first epoch), all tasks seem to manifest positive and similar inter-task affinity. We postulate the representations learned by the model at this early stage may be common among all tasks (i.e. honing in on where facial characteristics appear in an image), but specialization quickly occurs thereafter. Investigating these dynamics, as well as possibly developing a routing network to dynamically adjust sharing among tasks as a function of inter-task affinity transfer, is left to future work.
> 2. **Task affinities robustness:** Another excellent question -- our empirical findings indicate inter-task affinity scores between two tasks are maintained even when other tasks are switched in and out on the CelebA and Taskonomy datasets (see https://sites.google.com/view/anon-response-to-reviewer-1wzl for an example on CelebA). However, this is not guaranteed to always be the case. For instance, one task may cause significantly decreased model performance, or even divergence. In either event, it is likely that inter-task affinities will differ between cases. We will add this discussion on robustness to the paper.
> 3. **Distribution of random groupings and best/worst groupings:** Looking at the distribution of random groupings is interesting, and we will include these plots in the final version. As you might expect, the distribution is Gaussian, with the median closely approximating the mean, for the 2-split and 3-split random groupings in both CelebA and Taskonomy (shown at https://sites.google.com/view/anon-response-to-reviewer-1wzl. We will try to determine the distribution of the 4-split CelebA task groupings; although storing all possible 64-bit floating point values in memory to plot the distribution creates difficulties. The optimal and worst values were derived from an exhaustive search.

---

### Official Review · Reviewer_suuP · 2021-07-17

**Rating:** 8
**Confidence:** 5

**Summary:**

Authors propose a single run training method to find out what group of tasks should be learnt together based on the network capacity and computational budget. They evaluate TAG on CelebA and Taskonomy datasets.

**Limitations And Societal Impact:**

Yes, they have mentioned and analyzed very carefully in sec 5.2

**Main Review:**

The paper is well written and easy to understand.

The proposed method solves the task grouping problem very efficiently, which is a very challenging topic in MTL. Previous methods require training many networks to determine the task group while this method only requires a single run, which is very innovative and exciting. I like the method.

The method accumulates training gradients of different tasks during the single run and computes the task affinity, which sheds some light on the challenging task affinity computation. It is also very insightful in transfer learning.

In experiment, they evaluate on CelebA and Taskonomy datasets. TAG yields the best relative runtime and total test loss in both scenarios. They also carefully analyze the drawback of their method in Sec5.2 and also carefully discuss the frequency of computing the gradient and which data split should be used.

For Rebuttal,
It would be very interesting that the authors could provide some grouping examples, and show the difference when it is 2 groups, 3 groups and 4 groups. Task affinity matrix will also demonstrate the effectiveness of the method.



**Time Spent Reviewing:**

4hr

---

> ### Author Response · Authors · 2021-08-08
> **Response to Reviewer suuP**
>
> Thank you for your review! We are both honored and encouraged by your esteem for the work. We agree that visualizing some of the task grouping splits from Tables 5, 6, 7, 9, 10, and 11 of the Appendix would add to the paper and highlight the differences by varying the inference-time budget. Further, the suggestion of including a task affinity matrix to visualize inter-task affinity among tasks is extremely relevant and a compelling method to visually depict affinity among tasks. We present both visualization at https://sites.google.com/corp/view/anon-response-to-reviewer-suup (website analytics turned off to maintain the anonymity of the reviewer) and will include them in the final version.

---

> > ### Comment · Reviewer_suuP · 2021-08-22
> > **Thanks for authors' responses**
> >
> > I am grateful for the authors' response. They address my suggestions and concerns. I will keep my initial positive rating.

---

### Official Review · Reviewer_swCN · 2021-07-17

**Rating:** 8
**Confidence:** 4

**Summary:**

This paper proposes a method for grouping tasks to achieve stronger multi-task learning (MTL) performance. The approach is relatively simple and quite efficient in comparison to alternatives: during a single pass of learning, measure the impact that one task's gradient updates would have on other tasks. A theoretical analysis is conducted in a simplified (strongly convex) setting, and experiments show that this approach is as effective as alternatives that are much more computationally intensive. Overall this appears to be a useful technique that could be useful for many other researchers and also potentially for product teams.


**Limitations And Societal Impact:**

Limitations are not discussed. Societal impact is reasonable.


**Main Review:**

Strengths:
1. The approach is relatively simple, easy to understand, and (relatively) easy to implement.
2. The approach appears highly effective in the setting where the number of parameters total (across all models) is not constrained.
3. The experiments are conducted well, and the results are convincing.

Weaknesses:
1. While the paper makes an argument on the basis of deployed systems that it is reasonable to allow the number of parameters to grow as a function of the number of task groupings, this does introduce a confounding effect in analyzing the results. It also was unclear to me whether, for the comparison systems, if they were also allowed to have increasing numbers of parameters for the number of groups.
2. The assumption of convexity in the theoretical analysis, while probably the only way to make the result go through, is fairly unsatisfying in MTL settings, especially when the whole motivation is to learn good shared representations.



I only have a few additional questions/comments:

 - I would avoid the term "co-training" -- this is its own field of ML, quite different from MTL, and overloading the term is confusing.

 - I was confused throughout much of the paper about why the paper claims there are 2^T-1 possible groupings; I thought it should be B_T (the T-th Bell number). It turns out the answer is because tasks can contribute to the training of networks that do not make predictions for that task. This should be made clearer much sooner (eg around line 38).

 - l.103 "combined task performance" -- I assume this means "average task performance" (later text confirms this) but it would be good to be specific.

 - I find the terminology "inference-time budget" confusing because it's not clear what it's a budget over. I would call it "parameter budget" or "memory budget" or something like that.

 - l.107 I don't understand what "are not served" means.

 - In the theoretical analysis (4.3) it wasn't clear to me how this interacts with the affinity averaging described around line 168. It would be nice if this were clarified.

 - I'm very confused by Figure 2. My understanding is that what is plotted here is the runtime required to find groups (this is also what the caption says). But then why is STL 8.5x and 3.8x, respectively? There is no time required to find groups in STL. Similarly, why does "Full MTL" have a non-zero amount of time to "find groups"? With those as baselines, I would expect everything else to be infinitely longer. I'm clearly not understanding what is being plotted here, but l.237 syas "we report the time to determine task groupings, not determine task groupings and train the resultant multi-task networks."

 - I'd ideally like to see figures showing the full training time (including groupings and training) for all results. The proof of the theorem can be moved to the appendix to make space for this. I understand that the point of this paper is a method to find good groupings, but given that the paper stresses the practical relevance of this approach, what practitioners will really care about is the full time.

 - l.246 I don't understand the discussion of 2-splits, 3-splits, 4-splits here. I assume that this means that you're allowed to have 2* as many parameters, 3*, and 4*? Is that right? Please clarify.

 - Table 3 body and caption says "b=128x" but the text on l.346 and l.355 says "b=0.5x". Which is it?



**Time Spent Reviewing:**

1.5

---

> ### Author Response · Authors · 2021-08-08
> **Response to Reviewer swCN**
>
> We appreciate your careful review! You raise several salient points and  addressing them will improve this work. We address your questions/comments below:
>
> 1. **Are comparison systems allowed to increase in parameters with the number of groups:** The results in Figure 2 for optimization methods (UW, PCGrad, GN, etc.) used a single, consistent model architecture, but Figure 4 of the Appendix displays our experimental findings after removing the inference-time latency requirement for optimization methods and increasing the number of parameters to equal those used in task groupings.
> 2. **Overloading the term "co-training":** We agree and have replaced each of the 9 instances of the word co-training with appropriate substitutes.
> 3. **Number of possible groupings:** This is a good point. We have addressed this ambiguity in the introduction to detail the possibility of tasks contributing to the training of a network, but not making serving-time predictions.
> 4. **Combined task performance:** Your assumption is correct. We have corrected this ambiguity to clarify that combined task performance is equivalent to average task performance.
> 5. **Inference-time budget terminology:** Thank you for raising this point. We will replace "inference-time budget" with "memory budget/constraint".
> 6. **Are not served terminology:** The phrase "are not served" was intended to specify the tasks that contribute to the training of a particular network; however at serving, the predictions for these tasks are discarded or not computed. You touch on this concept in bullet point 2 with "tasks can contribute to the training of networks that do not make predictions for that task". We have rephrased and clarified this point in the text.
> 7. **Link between averaging inter-task affinities and theoretical analysis:** While our theoretical analysis considers the effect of a single task on another, its generalization to a set of tasks involves replacing the single-task gradients with the sum of the gradients over the set of tasks in a group. Therefore, our results will hold exactly if we define the task affinity measure as a linear combination of the task gradients. However, this is computationally prohibitive as it requires calculating the change in loss for all possible combinations of tasks. Averaging pair-wise affinity scores seems to be a tractable approximation that also works well in practice. We will further clarify this link and expand our analysis.
> 8. **Figure 2 Ambiguity:** In Figure 2, we reported only the time to determine task groupings for task grouping methods (TAG, CS, HOA) and the training time of non-task grouping methods (MTL, GN, PCGrad, UW). We wanted to contextualize the time to find task groupings relative to the typical training time of popular MTL methods (MTL, UW, PCGrad, etc.). We will expand our description from L237-241, as well as the Figure 2 caption, to better convey this distinction. Additionally, we will include a figure (see next bullet point in our response) showing the full run time (time to determine task groupings + train resultant networks) for all task grouping results.
> 9. **Full training time figure:** As practitioners ourselves, we wholeheartedly agree with this statement. We present this figure at https://sites.google.com/view/anon-response-to-reviewer-swcn (website analytics turned off to maintain the anonymity of the reviewer), and will include it in the final version.
> 10. **2-split, 3-split, 4-split parameter ambiguity:** You are correct; the 2-split, 3-splits, and 4-split groups have 2*, 3*, and 4* the number of parameters as the optimization methods but the inference-time latency is identical when run in parallel. Alternatively, increasing the number of parameters within a single model may significantly increase the time to make a forward pass during inference. Figure 4 of the Appendix contains our findings when we remove the inference-time latency constraint from comparison systems.
> 11. **b=128x mistake:** Great catch. b=128x is an artifact from an early draft, signifying a batch size of 128 (half the size of the typical setting). We have updated our draft to fix this error (b=128x -> b=0.5x).

---

> > ### Comment · Reviewer_swCN · 2021-08-28
> > **Thanks for the clarifications**
> >
> > Thank you for the clarifications. Everything sounds great except I still don't understand what is being plotted in Figure 2, and in particular what the time to find grouping for STL means. This isn't going to affect my score, but I hope you can find a way to make it clear!

---

> > > ### Author Response · Authors · 2021-08-29
> > > **Follow-Up Response to Reviewer swCN**
> > >
> > > We sincerely appreciate you leaving this final note. The runtime for STL in Figure 2 is the runtime to train all 9 single task models. We were mistaken to not differentiate this method from the task grouping methods (TAG, CS, HOA).
> > >
> > > Given your comments, we now see that comparing the training times of non-task grouping methods (MTL, GN, PCGrad, STL, etc.) with the time to find groupings for task grouping methods (TAG, CS, HOA) can be ambiguous and confusing. We commit to making these results more clear by reporting the time to find task groupings + time to train the resulting groups in Figure 2 relative to the time to train a single MTL model. Including the Figure we created to address bullet point 9 of your initial review (https://sites.google.com/view/anon-response-to-reviewer-swcn) will showcase the difference in time to find groupings among task grouping methods, the information we initially sought to highlight in Figure 2.

---

### Author Response · Authors · 2021-08-08
**Meta Response**

We thank the reviewers for their insightful and positive feedback! We are encouraged they find the approach straightforward (swCN, 1wZL, LXec), highly effective (swCN, suuP, 1wZL), and relevant (swCN, suuP, 1wZL). Several points raised by the reviewers suggest changes to the text to improve the overall coherence and lucidity, while others recommend additional figures and visualizations to further buttress the analysis. We answer some of their specific comments in our response, and while this venue does not permit the posting of paper revisions during the authors' response, we will incorporate all feedback into the final version.

---

### Decision · Program_Chairs · 2021-09-27

**Decision:**

Accept (Spotlight)

**Comment:**

This paper proposes a method for grouping tasks to achieve stronger multi-task learning performance.  All four reviewers believe this is a well-written paper and should get accepted.   Authors need to modify their final version based on reviewers' comments.